# Immunohistochemical Expression of Tensin-4/CTEN in Squamous Cell Carcinoma in Dogs

**DOI:** 10.3390/vetsci10020086

**Published:** 2023-01-24

**Authors:** Alexandra Monteiro, Leonor Delgado, Luís Monteiro, Isabel Pires, Justina Prada, Teresa Raposo

**Affiliations:** 1Department of Biology and Environment, University of Trás-os-Montes and Alto Douro, 5000-801 Vila Real, Portugal; 2Instituto de Investigação e Inovação em Saúde, Universidade do Porto (i3S), 4200-135 Porto, Portugal; 3UNIPRO-Oral Pathology and Rehabilitation Research Unit, University Institute of Health Sciences-CESPU (IUCS-CESPU), 4585-116 Gandra, Portugal; 4Pathology Department, INNO Serviços Especializados em Veterinária, 4710-503 Braga, Portugal; 5Department of Veterinary Science, University of Trás-os-Montes and Alto Douro, 5000-801 Vila Real, Portugal; 6CECAV-Veterinary and Animal Research Center, University of Trás-os-Montes and Alto Douro, 5001-801 Vila Real, Portugal; 7Horizon Discovery, Cambridge CB25 9TL, UK

**Keywords:** CTEN, SCC, tumor progression

## Abstract

**Simple Summary:**

Tensins are a family of focal adhesion proteins that connect the extracellular matrix (ECM) to the cytoskeletal networks, mainly through integrin receptors and their associated protein complexes. To investigate, for the first time in canine species, the expression of CTEN was evaluated in oral and cutaneous squamous cell carcinoma samples from several dog breeds and submitted to immunohistochemistry technique. The results reveal a higher level of expression associated with the highest grades of the tumor, evidencing that this protein is more expressed in the highest grades of squamous cell carcinoma, probably due to its involvement in tumor progression. In the present study, we demonstrated, for the first time, CTEN clinical significance in a canine squamous cell carcinoma, and, to our knowledge, this is also the first time that the CTEN expression pattern is linked to the canine species.

**Abstract:**

C-terminal tensin-like (tensin-4/TNS4/CTEN) is the fourth member of the tensin family, frequently described as displaying oncological functions, including cellular migration, invasion, adhesion, growth, metastasis, epithelial to mesenchymal transition, and apoptosis, in several different types of cancer. To investigate, for the first time, the clinical significance of CTEN in squamous cell carcinoma (SCC) of dogs, we studied a total of 45 SCC sections from various dog breeds. The mean age of the affected dogs was 8.9 ± 3.6 years. Immunohistochemistry confirmed strong cytoplasmatic CTEN expression in the basal layer of the epidermis next to the tumor. We detected high CTEN expression associated with the highest grade of the tumor (grade III) and observed 100% of immunopositivity for this tumor grading (*p* < 0.0001). These data suggest that CTEN is an oncogene in SCC of dogs and a promising biomarker and a therapeutic target for dogs affected by SCC.

## 1. Introduction

C-terminal tensin-like (tensin-4/TNS4/CTEN) was the last member identified in the tensin family [1]. It was described as a much smaller protein (80 kDa), having in common with other tensins (tensin-1, tensin-2, and tensin-3) its C-terminal (carboxyl-terminus), containing both phosphotyrosine binding (PTB) and Src homology 2 (SH2) domains [2]. These domains are extremely important for the smooth operation of all tensins, despite some of their functions being conflicting between them. The SH2 domain plays a major role in tensin’s signaling pathways since it binds with phosphorylated proteins in their tyrosine amino acid, such as phosphoinositide 3-kinase (PI3K), focal adhesion kinase (FAK), tyrosine-protein kinase MET (MET), Crk-associated substrate (p130CAS), paxillin, epidermal growth factor receptor (EGFR), casitas B-lineage lymphoma (c-Cbl), proto-oncogene tyrosine-protein kinase (Src), and tyrosine-protein kinase receptor UFO (AXL) [3,4,5,6]. The PTB domain is also important for tensins since it binds with ꞵ-integrin at the NPXY/F motifs and controls the signaling pathways that regulate cell adhesion and motility [6].

Despite tensins being known to localize to focal adhesions, CTEN is also found in the cytoplasm and the nucleus, and it is in these three sub-cellular localizations that CTEN exhibits its physiological functions [7,8]. Being positively regulated by several growth factors, including epidermal growth factor (EGF), transforming growth factor-beta 1 (TGF- β1), insulin-like growth factor 1 (IGF-1), interleukin-6 (IL-6), interleukin-13 (IL-13), and fibroblast growth factors (FGFs), CTEN is responsible for controlling multiple proteins and signaling pathways, through which CTEN regulates cell invasion, migration, adhesion, growth, metastasis, apoptosis, and epithelial to mesenchymal transition (EMT) [3,4,5,7,9,10,11,12,13,14,15,16,17]. Bearing that in mind, it is not surprising that CTEN is frequently reported as an oncogene in human cancers, although it was initially described as a tumor suppressor in prostate cancer, with very limited expression in normal tissues [1]. Overexpression of CTEN was reported in several human cancers, such as thymoma [18], melanoma [19], hepatocellular carcinoma [20], lung [13], colon [12,21], breast [10], pancreatic [22], ovarian [16], and gastric [23]. These reports prove that CTEN contributes to carcinogenesis, being also widely considered to be a putative biomarker and a therapeutic target, although its role and its expression are poorly understood, especially in veterinary medicine. As far as we know, only two studies were published using MDCK (Madin–Darby canine kidney) cells examining the mechanotransduction pathway of TNS4 and the keratin network [24], and the role of the TNS4-STAT3 axis in epithelial sheet invasion and tubulogenesis [25]. However, there are no studies on its role in cancer in dogs. 

Squamous cell carcinoma (SCC) is a malignant tumor derived from epidermal keratinocyte differentiation [26], commonly due to DNA damage, induced by UV radiation [27]. SCC can affect not only humans, but also dogs, cats, cattle, caprine, and ovine species, affecting many different body regions, including the digits, lips, eyes, skin, lungs, anus, abdomen, ears, lips, mammary gland, oral cavity, and nasal cavity [28,29,30]. Canine SCC is a very common cancer disease in dogs (up to 2% of all cancers) [31]. The risk for the development of SCC is associated with increased age, exposure to solar radiation, and lack of haircoat/pigment within the epidermis [32], predominantly affecting the digital region, oral cavity, and skin [30,32]. SCC is often described as a locally invasive carcinoma with a low capacity for slow metastases [33], but this may vary depending on the localization of the tumor. 

Comparative oncology has gained increasing interest in the scientific community. Studies in animal tumors can open avenues for studies in human tumors and vice versa. Given the percentage of identity between dogs’ (ENSCAFT00845014125.1) and humans’ (ENST00000254051.11) CTEN protein (up to 84%) [34], it is possible that CTEN may display similar functions in both species. Furthermore, besides the high homology shared between them, their physiological and histological processes can be very similar [35]. Thus, we suspect that CTEN may have an important role in canine SCC carcinogenesis, since it was recently reported to be overexpressed and implicated in the repression of apoptosis in human squamous cell carcinoma of the head and neck [14].

With this study, we intended to evaluate CTEN expression in squamous cell carcinoma of dogs with different ages, breeds, and tumor localizations, describing for the first time not only CTEN immunoreactivity in SCC but also its pathological involvement in canine tumors.

## 2. Materials and Methods

### 2.1. Animals and Histopathology

A total of 45 canine SCC histological sections were obtained from the Histopathological Laboratory of the University of Trás-os-Montes and Alto Douro archives. These tumors were excised from 45 dogs and previously fixed in 10% buffered formalin and paraffin-embedded. For each case, clinical information was recorded such as gender, age, breed, and localization (oral or cutaneous) and size of the tumor. It was not possible to obtain the clinical staging or follow-up of animals included in the study.

For the histopathological study, sections of tissue 4 µm thick were stained with hematoxylin and eosin. Two independent pathologists reviewed each sample. All slides were considered in this study and entire sections of the tumors were analyzed.

Histopathologic diagnosis was based on the World Health Organization (WHO) histological classification of animal tumors [36].

A Nikon Eclipse E600 microscope with a Nikon DXM1200 digital camera (Nikon Instruments Inc., Melville, New York, USA) was used for microscopical observations and images capture. 

### 2.2. Histopathological Evaluation

Histological grading was assigned using an adapted version of Anneroth’s multifactorial system [37]. The grading was based on the histological classification of the malignity of tumor cells, scoring three different parameters from 1 to 3, including the degree of keratinization, nuclear polymorphism, and the number of mitoses. The degree of keratinization was accessed according to the percentage of tumor keratinized cells into: I—>50% cells keratinized; II—20–50%; and III—0–20% cells keratinized. Nuclear pleomorphism was evaluated as: I—little (> 75% mature cells); II—moderate (50–75% of mature cells); III—abundant nuclear pleomorphism (0–50% of mature cells). The number of mitoses was counted in 10 high power field (HPF) and categorized as: I—0 to 1 mitosis/HPF; II—2 to 3 mitosis/HPF; III—≥ 4 mitosis/HPF.

The tumor–host relationship was also evaluated, depending on the pattern of invasion, stage of invasion, and lymphoplasmacytic infiltration, scoring each parameter from 1 to 3. The pattern of invasion was considered as follows: I—pushing, well-delineated infiltrating borders; II—infiltrating solid cords, bands, and/or strands; III—small groups, cords, or single cells infiltrating. Stage of invasion was graded as: I—corresponding to carcinoma in situ and/or questionable invasion; II—distinct invasion, only involving *lamina propria*; III—invasion below *lamina propria*, including muscles. Lymphoplasmacytic infiltration was evaluated as: I—marked; II—moderate; III—slight to none.

The total points calculated were divided into three grades: grade I (score 5–10), including well-differentiated tumors; grade II (score 11–15), including moderately differentiated tumors; and grade III (+16), including poorly differentiated tumors.

### 2.3. Immunohistochemical Staining Procedure

The immunohistochemical (IHC) technique was performed in 45 canine SCC sections (3 μm thick) from formalin-fixed paraffin-embedded (FFPE) tissues to investigate CTEN’s expression. The protocol was divided into two parts and included the usage of the kit NovolinkTM Polymer Detection Systems (Leica Biosystems^®^, Newcastle, UK).

In the first part, the samples were dewaxed in xylene for 15 min, followed by hydration in graded alcohols (100%, 95%, 80%, and 70%), for 5 min each. For antigen retrieval, the samples were submersed in citrate buffer solution (10 mM, pH 6.0 ± 0.2) and heated in a 750 W microwave for 3 cycles, 5 min each. For peroxidase inhibition, we applied 3% hydrogen peroxide (H_2_O_2_) for 30 min and, after washing with PBS, the sections were incubated with Protein Block for 5 min. The first part of this procedure ended with the incubation of the antibody tensin-4 (Thermofisher^®^, Waltham, MA, United States), clone SP83, rabbit anti-human, Invitrogen MA516355), diluted 1:100 in PBS for 24 h at 4 °C. In the second part of the protocol, we applied post-primary reagent, followed by Novolink Polymer, each for 30 min. After several washings with PBS, the samples were revealed, by incubation of 3,3-diaminobenzidine (DAB) (Novocastra^®^, kit NovolinkTM Polymer Detection Systems, Leica Biosystems^®^, Newcastle, UK) for 10 min and counterstained with Gill’s hematoxylin for 1 min. After washing with water, the sections were dehydrated in graded alcohols (95%, 95%, 100%, 100%) for 3 min each, diaphanized in xylene, and then mounted with the aqueous mounting medium Entellan (Merck^®^, Rahway, NJ, United States).

We performed two different types of control tissues, namely positive and negative control. As a positive control, we chose a tissue that was collected from a canine prostate tumor, since CTEN expression in normal tissues is restricted to the prostate and placenta in humans [1]. For the negative control, one SCC sample was treated with PBS instead of using the antibody, to demonstrate that the staining was a result of the interaction between the target cell and CTEN.

### 2.4. Immunoreactivity of CTEN

The immunoreactivity was qualitatively analyzed and classified as positive or negative, depending on the staining intensity, when compared with the basal layer of the epidermis next to the tumor, since CTEN expression was observed to be significantly higher in this region, as it was described in human skin [38]. The samples were classified as positive when the staining intensity was stronger than the basal layer of the epidermis next to the tumor, and negative when the staining intensity was similar or lower.

### 2.5. Data Analysis

For statistical data analysis we used SPSS (Statistical Package for the Social Sciences, IL, EUA) software, version 24.0, in which we performed Pearson’s chi-squared test (ꭓ2) and Fisher’s exact test. The results were considered statistically significant for *p* < 0.05.

## 3. Results

### 3.1. Clinical Information

Overall, 48.9% (22 cases) were female and 42.2% (19 cases) were male, having missing data from four animals. The ages ranged from 1 to 17 years (mean = 8.9 ± 3.6 years) and their breed details are given in Table 1. Most of the SCC samples were from the oral cavity (50%), and the others were from several parts of the *corporis integument* (50%) [39]. The tumor sizes were larger than 2 cm in 66.7% of the cases.

### 3.2. Histological Classification of the SCC Tumors

The classification was attributed according to the criteria that were described above: 33.3% (15 cases) were well-differentiated (grade I); 37.8% (17 cases) were moderately differentiated (grade II); 28.9% (13 cases) were poorly differentiated (grade III); Figure 1.

### 3.3. Immunoexpression of CTEN

The expression of CTEN was observed in the basal layer of the non-tumoral epidermis next to the tumor and it was cytoplasmatic. In tumor cells, the labeling was cytoplasmatic, being noted mainly in the areas of invasion (Figure 2).

In total, 55.6% (25 squamous cell carcinomas) revealed negative CTEN expression and 44.4% (20 cases) were positive for this marker. Considering the well-differentiated tumors, only 5% (1 case) showed positive CTEN expression, while 93.3% (14 cases) accounted for negative CTEN expression. Moderately differentiated squamous cell carcinomas (*n* = 17) revealed negative CTEN expression in 11 cases (64.7%) and a positive reaction in 6 cases (35.3%) (Figure 3). The higher tumor grading (poorly differentiated tumors) showed CTEN immunoreactivity in all cases (*n* = 13; 100%); Figure 4.

Analyzing the association of CTEN immunoreactivity and the histological grading of the tumors, a significant association of the higher tumor grading (poorly differentiated tumors) with CTEN immunoreactivity was noted, which was statistically significant (*p* < 0.0001).

A possible association between CTEN expression and the localization of the tumors (cutaneous/oral) was also evaluated. However, the differences between CTEN expression in cutaneous and oral tumors were not statistically significant (*p* = 1); Figure 5.

## 4. Discussion

CTEN was first described to have restricted expression in the human prostate and placenta [1]; however, after almost 20 years of studying this protein and its gene, it has been frequently reported to be upregulated in multiple human cancers, suggesting its role as an oncogene. In animals, in vitro studies using Madin–Darby canine kidney (MDCK) cell lines suggested that CTEN is involved in multiple cellular mechanisms, including invasion of the extracellular matrix, through CTEN–keratin network interaction, since keratins are key factors in cell migration [24,25]. However, there is no evidence regarding CTEN expression in canine tumors; therefore, our main aim was to evaluate CTEN immunoexpression in canine SCC.

A total of 45 SCC tumor samples with different localizations and histological gradings, from dogs with distinct breeds (Table 1), genders, and ages were studied. The mean age obtained was 8.9 ± 3.6 years old, matching the values found in the literature [30,32,40,41]. The oral cavity was representative of 50% of the SCC samples, and the other 50% were from different parts of the *corporis integument* [39]. We could not find any association between the histological grading of the tumors and their localization (*p* = 1).

The immunohistochemical procedure revealed a cytoplasmatic staining, more predominant in the basal layer of the epidermis next to the tumor, corroborating the findings of Seo and his colleagues, in human skin [38]. They suggested a possible role for CTEN as a signal adapter, in epidermal proliferation and/or differentiation, transmitting integrin signals from integrin subunit beta 4 (ITGB4) to the FAK/ERK signaling pathway [38]. Thereby, in our experiment, in the absence of any reports in canine tumors and after having the same staining pattern as those found in human skin, we considered positive labeling when the staining in SCC tumor sections was stronger than in the basal layer of the skin next to the tumor.

Our results show 44.4% positive immunoreactivity for CTEN, suggesting a strong concordance with the results found in head and neck squamous cell carcinoma in humans, which reported different levels of expression between normal tissues and SCC samples of head and neck tumors [14]. In squamous cell carcinomas of dogs, mechanisms associated with cell survival similar to those described in humans may be implicated and should be explored. In human head and neck SCC, CTEN was found to regulate several apoptotic proteins, including the 60 kDa heat shock protein (HSP60) and caspase-3, which are downregulated by CTEN, and conversely, upregulated proteins, including the Bcl-2-like protein 1 (BCL2L1) and myeloid cell leukemia-1 (MCL1) [14]. Others have demonstrated a role in inducing statin-induced apoptosis resistance, increasing the invasive and metastatic potential of tumor cells [9], and a direct interaction between the SH2 domain of CTEN and MET, which reduces its endocytosis and increases cell survival potential [16]. These and other biomarkers that have been shown to be associated with CTEN in human tumors could be an interesting and promising field of study in canine tumors.

Another interesting finding reported in the present study is the positive correlation between the histological grading of the tumors and CTEN expression, evidencing the aggressiveness of this marker. The highest histological grade (grade III) revealed 100% of immunopositivity (*p* < 0.0001). A similar pattern of results was obtained in different human cancers, such as adenocarcinoma of the esophagogastric junction [42], thymoma [18], lung [43], gastric [23], and breast cancer [10]. This evidence combined suggests a strong involvement of CTEN in cancer progression, which is involved mainly in cell invasion and metastasis. Additionally, this highlights CTEN’s potential as a biomarker and a therapeutic target in a canine SCC.

The physiological functions displayed by CTEN contributing to cancer progression in canine SCC can probably have similar mechanisms to those reported in other human cancers. Induced EMT is described in many human cancers, and during this process epithelial cells change their polarity, reorganize cytoskeletal systems, and lose their cell-to-cell adhesion, involving changes in the pattern of expression of some proteins (e.g., E-cadherin and N-cadherin), conferring motility (both invasion and migration) to tumor cells [44]. Downregulation of E-cadherin, despite being linked to a shift of E-cadherin to N-cadherin during the EMT process [45], has been highly associated with upregulation of CTEN in human cancers, CTEN being considered a mediator in integrin–cadherin crosstalk, promoting E-cadherin inhibition through its degradation or post-transcriptional repression [22,46]. Indeed, in colorectal cancer (CRC) cell lines, TNS4 has been shown to induce EMT by repressing E-cadherin at the post-transcriptional level and significantly increase migration and invasion [46]. Nuclear localization of TNS4 has been previously identified in colorectal cancer metastases [9], and it is likely that having undergone EMT, these metastases would express low E-cadherin. Induced EMT was associated with the invasive profile of canine cutaneous and oral SCC, in which the expressions of E-cadherin and β-cadherin were significantly low, associated with the invasive front of the histological grade [47]. It is possible that CTEN may enhance cellular invasion in canine SCC, through the downregulation of E-cadherin and β-cadherin in the cytoplasmic membrane. Other pathways can be implicated in CTEN signaling to promote EMT, including upregulation of SNAIL [17] or induction of the signaling pathway Src/ROCK/SNAIL, promoted by TGF-β1, an upstream regulator of CTEN [11]. Cell migration and invasion have been proven to be important roles of CTEN, involving several proteins and signaling pathways, including KRAS/BRAF, Src/ROCK/SNAIL, ILK, and FAK in colorectal and pancreatic cancer [12,21,22], and Rho/ROCK/MLC in hepatocellular carcinoma, via DLC-1 [48,49]. The metastatic and invasive potential can also be enhanced by invasion into the ECM, induced by HGF [50] or through regulation of adhesiveness to collagen I [51].

Interestingly, in canine spontaneous head and neck SCCs similar alterations were found to those described in human head and neck SCC, including dysregulation in the PI3K/AKT/mTOR and TGF-β signaling pathways, and in EMT, through increased expression of vimentin and diminished expression of E-cadherin [52]. In canine cutaneous SCC, PI3K/AKT/mTOR was also found to be activated by immunodetection of the phosphorylated forms of the proteins AKT and S6 [53]. Overexpression of phospho-S6 and phospho-mTOR were also detected in canine oral SCC [54]. However, these mechanisms are not clarified in canine SCC, and the role of CTEN in dog pathology, so far, has never been investigated, so all of these possible interactions are just hypotheses that need to be explored. Indeed, in future studies, the expression of CTEN could be evaluated together with EMT markers, in canine SCC, to clarify the mechanism with which CTEN induces invasion in these tissues. Additionally, it could be interesting to evaluate the role of CTEN in the regulation of apoptosis in canine SCC, so as to explore if the findings described in human head and neck squamous cell carcinoma can be compared to canine SCC. Translational research is a promising strategy for better understanding cancer biology, therefore improving the treatment options and diagnostic methodologies available in both human and veterinary medicine. For instance, companion animals, such as dog species, are genetically very close to human species, share the same environment with humans, have similar clinicopathological features, and their shorter life span makes them good candidates for studying human diseases in naturally occurring animal models [35].

We acknowledge some limitations of the present work, mainly related to the retrospective nature of the study, the relatively small number of cases, and the lack of some clinical information. Nevertheless, we investigated, for the first time, CTEN’s behavior in canine SCC, and based on the correlation of positive immunoreactivity and increasing histologic grade in this study, we postulate that CTEN may be involved in tumor progression.

## 5. Conclusions

To our knowledge, we have reported for the first time CTEN expression in canine SCC. The results from our experiment reveal a higher level of expression associated with elevated invasive behavior and higher histological grade of the tumor, evidencing that this protein is more expressed in poorly differentiated squamous cell carcinomas, probably due to its involvement in tumor progression.

Overall, our results suggest that CTEN contributes to tumor progression in canine SCC, therefore revealing CTEN’s potential as a biomarker and a therapeutic target for canine SCC, as well as its possible value in histological grading of invasive fronts. However, future studies are required to focus on the protein networks displayed by CTEN and to clarify the mechanisms associated with CTEN in the carcinogenesis of squamous cell carcinoma.

## Figures and Tables

**Figure 1 vetsci-10-00086-f001:**
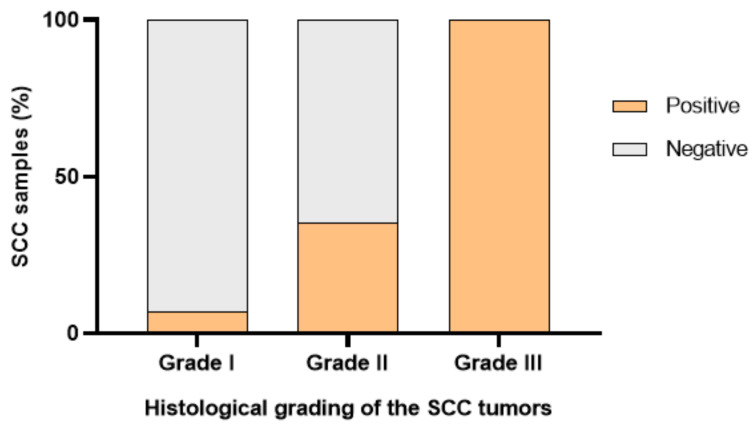
CTEN immunoreactivity (positive and negative) in grade I, grade II, and grade III SCC.

**Figure 2 vetsci-10-00086-f002:**
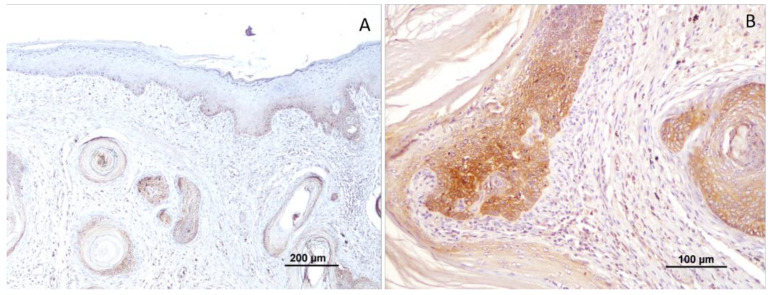
CTEN immunoreactivity in well-differentiated tumors (grade I). (**A**) Negative immunoreactivity. Note the non-tumoral epidermis with labeling similar to tumoral cells. (**B**) Positive immunoreactivity.

**Figure 3 vetsci-10-00086-f003:**
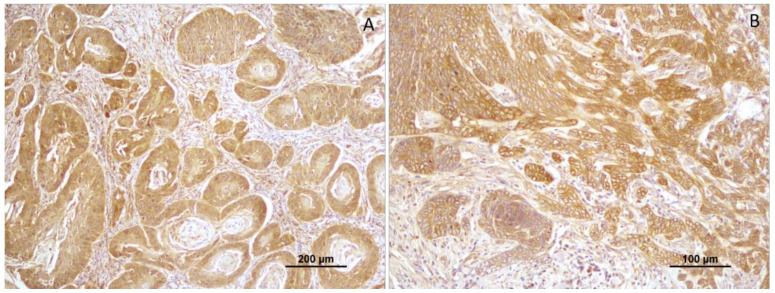
(**A**, **B**) Positive CTEN immunoreactivity in moderately differentiated tumors (grade II).

**Figure 4 vetsci-10-00086-f004:**
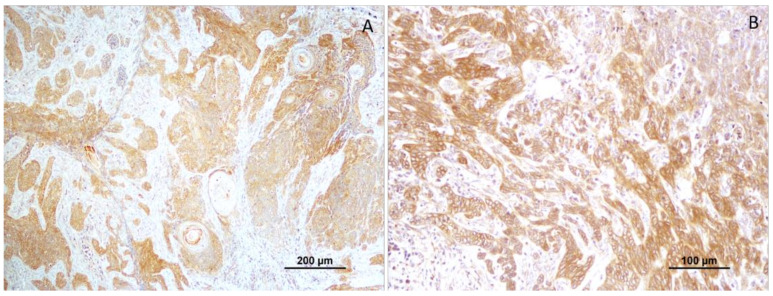
(**A, B**) Positive CTEN immunoreactivity in poorly differentiated tumors (grade III).

**Figure 5 vetsci-10-00086-f005:**
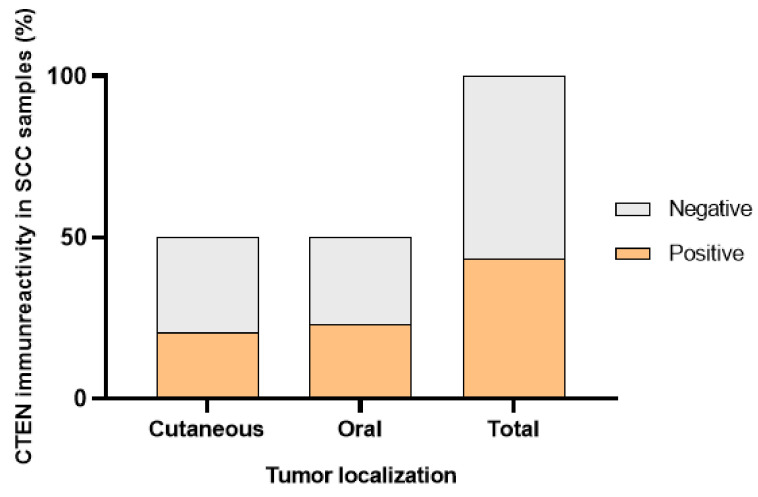
CTEN immunoreactivity in cutaneous and oral SCC.

**Table 1 vetsci-10-00086-t001:** Percentage and frequency of dog breeds with SCC.

Breed	Frequency	Percentage
Yorkshire	2	4.4
Boxer	2	4.4
Basset Hound	3	6.7
Bobtail	1	2.2
West Highland terrier	1	2.2
Poodle	3	6.7
Castro laboreiro	2	4.4
Cocker Spaniel	1	2.2
Border Collie	1	2.2
Dalmatian	3	6.7
Golden Retriever	1	2.2
Siberian Husky	2	4.4
Labrador Retriever	2	2.2
Portuguese Pointer	1	2.2
Pitbull	1	2.2
Podengo	1	2.2
English Pointer	1	2.2
Spanish Scenthound	1	2.2
Giant Schnauzer	1	2.2
Estrela Mountain Dog	2	4.4
Irish Setter	1	2.2
Sharpei	1	2.2
Mixed breed	8	17.8
Unknown	3	6.7
Total	45	100

## Data Availability

The data information can be asked for from the corresponding author.

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
