# Peer review of "Immunohistochemical Expression of Tensin-4/CTEN in Squamous Cell Carcinoma in Dogs"

_vetsci, 2023, doi:10.3390/vetsci10020086_

Round 1

Reviewer 1 Report

Source text

Dear Authors, the topic is interesting and thematic, but some important changes and refining must be operated for improving the article to make it suitable for publication.

The most important revision must be performed in the chapter of "Discussion". The follow sentences (from line 200 to line 205) are all related to a bibliography on studies performed in human beings transferring them sic et simpliciter to the canine species without having specific research  in the dog. It is necessary to have bibliographic citations of species-specific studies in dog to support the numerous regulatory/downregulation actions with an irrefutable approach referred in canine specie.

The "Conclusions" must be revised after "Discussion" revision.

The bibliography must be reduced to essential citations (from line 63 to line 69).

In "Materials and Methods" - the histological methods are not described or detailed ( es. thick of histological and IHC sections);

The Figure 2 is composed by 4 (four) pictures but the caption are referred only a 3 (three) that are those aligned in the top, but that do not have any indication of the reference letter (A, B, C) used in captions. The fourth picture (the one with the black arrows) does not have any caption and must be improved for having a clear definition of details. In this picture it is also useful to insert the micrometrical bar.

A histological pictures of the three different histological grading would be useful to insert them to implement the communicative effectiveness  of the article .

line 28 "companion animal tumor". The investigation has been performed only on canine species so I invite the authors to rephrase the sentences;

line 57 "subcellulare locations". It is better rewrite with the term sub cellular localization;

line 137 "having missing data of 4 sections". It is better rewrite the sentence changing the word sections with the word samples;

line 182 "companion animal tumor" same comment of line 28;

line 187 "cutaneous region". Nomina Anatomica Veterinary 2017 sits editions use the term Partes corporis of integument (link https://www.wava-amav.org/wava-documents.html).

It is advisable to rewrite the sentence as" in different partes of corporis integument" citing the bibliographic source.

Author Response

1- Dear Authors, the topic is interesting and thematic, but some important changes and refining must be operated for improving the article to make it suitable for publication.

The most important revision must be performed in the chapter of "Discussion". The follow sentences (from line 200 to line 205) are all related to a bibliography on studies performed in human beings transferring them sic et simpliciter to the canine species without having specific research in the dog. It is necessary to have bibliographic citations of species-specific studies in dog to support the numerous regulatory/downregulation actions with an irrefutable approach referred in canine specie.

            We appreciate your comments and have thoroughly researched the available literature for studies pertaining to the canine species and TNS4 expression. There are in fact two studies published using MDCK (Madin Darby Canine Kidney) cells examining the mechanotransduction pathway of TNS4 and the keratin network (Cheah et al., 2019 - PNAS) and the role of the TNS4-STAT3 axis in epithelial sheet invasion and tubulogenesis (Kwon et al., 2011 - Current Biology). These have now been added to the discussion.).

2- The "Conclusions" must be revised after "Discussion" revision.

The authors revised the conclusions.

3- The bibliography must be reduced to essential citations (from line 63 to line 69).

The authors corrected this.

4- In "Materials and Methods" - the histological methods are not described or detailed ( es. thick of histological and IHC sections);

            The information has now been added.

5- The Figure 2 is composed by 4 (four) pictures but the caption are referred only a 3 (three) that are those aligned in the top, but that do not have any indication of the reference letter (A, B, C) used in captions. The fourth picture (the one with the black arrows) does not have any caption and must be improved for having a clear definition of details. In this picture it is also useful to insert the micrometrical bar.

A histological pictures of the three different histological grading would be useful to insert them to implement the communicative effectiveness of the article.

            The authors added figures 2,3 and 4.

6- line 28 "companion animal tumor". The investigation has been performed only on canine species so I invite the authors to rephrase the sentences;

            We have changed this sentence to “this is also the first time that CTEN expression pattern is linked to the canine species.”

7- line 57 "subcellulare locations". It is better rewrite with the term sub cellular localization;

            We have amended subcellular locations to subcellular localization.

8- line 137 "having missing data of 4 sections". It is better rewrite the sentence changing the word sections with the word samples;

            We have changed sections for samples.

9 - line 182 "companion animal tumor" same comment of line 28;

            This term has now been removed.

10- line 187 "cutaneous region". Nomina Anatomica Veterinary 2017 sits editions use the term Partes corporis of integument (link https://www.wava-amav.org/wava-documents.html).

It is advisable to rewrite the sentence as" in different partes of corporis integument" citing the bibliographic source.

            We modified the sentence 

Reviewer 2 Report

This paper describes the enhanced expression of CTEN in canine squamous cell carcinoma in dogs. Expression data are convincing, but a bit unbalanced with only CTEN expression measurements and all kind of other proteins discussed lateron.

minor points:

 lines 46=48 strange sentence

summing up table 1 I come to 44 dog (having dyslexia I might be wrong)

IHC picture is difficult to see non-staining i n nucleus (see lines 55-57

is expression correlated to e.g. survival time?

The discussion is not focussed but contains all kind of relations with other proteins.  Is it possible to stain for some of these protein to give more body to this manuscript as now it only contains one observation

If E-cadherin is downregulated what do you expect for the localization of CTEN viz cytoplasmic or nuclear

Author Response

This paper describes the enhanced expression of CTEN in canine squamous cell carcinoma in dogs. Expression data are convincing, but a bit unbalanced with only CTEN expression measurements and all kind of other proteins discussed later on.

 The authors modified the discussion.

minor points:

1-  lines 46=48 strange sentence

The authors omitted the sentence

2- summing up table 1 I come to 44 dog (having dyslexia I might be wrong)

The authors correct the table. A Labrador retriever was missing,

3- IHC picture is difficult to see non-staining in nucleus (see lines 55-57

The authors have added new photos, according to the request, which also coincides with the request of other reviewers. Being the first paper of this biomarker in canine SCC, to the best of our knowledge, all immunoreactivity higher than that of the basal layer of the non-tumour epidermis was considered.

4- is expression correlated to e.g. survival time?

Unfortunately, it was not possible to obtain the clinical follow-up data of the animals. Our aim was not direct to survival analysis or to assess whether there is a relationship with survival but more direct to other clinicopathological factors as for example histological grade. On future research, probably in a prospective design, we intent to direct to survival analysis. We acknowledge this in the paper in material and methods and also in the limitations of the study.

5- The discussion is not focussed but contains all kind of relations with other proteins.  Is it possible to stain for some of these protein to give more body to this manuscript as now it only contains one observation

Our focus of the present work was on CTEN protein. We could include other proteins to comparison, but unfortunately, the blocks were used for other molecular techniques and some were destroyed. We ask for the reviewer's understanding. 

6- If E-cadherin is downregulated what do you expect for the localization of CTEN viz cytoplasmic or nuclear

In colorectal cancer (CRC) cell lines, TNS4 has been shown to induce epithelial to mesenchymal transition (EMT) by repressing E-cadherin at post-transcriptional level and to significantly increase migration and invasion (Albasri et al., 2009 - J.Pathol). Nuclear localization of TNS4 has been previously identified in metastases of colorectal cancer (Albasri et al., 2011 - Oncogene), and it is likely that having undergone EMT, these metastases would express low E-cadherin.   In our SCC samples, in case E-cadherin was downregulated, we would expect CTEN to have a nuclear localization, but we do not have data to corroborate this hypothesis.

Reviewer 3 Report

In the "Simple Summary", "Abstract", and within the body of the manuscript (page 7, line 244) the authors at times state that there is a correlation with CTEN positive staining and advanced stage of the tumor, when in fact it is a correlation with grade.

Is there further information on the clinical outcome of these patients, especially as correlated to the grading system?  Or any information regarding clinical staging at time of histopathology?  If not, I would explicitly state this in the manuscript.

In Section 3.4 "Immunoreactivity of CTEN" the use of percentages of positive and negative immunoreactivity for each grade does not seem consistent.  Based upon Figure 1 it seems that percentages should be reported for each tumor grade separately (and this would make most sense).  However, in the text it seems as if percentages are listed across all grades.  Please correct and use consistently.  A table would also be helpful to clarify this information.

The summary statement on page 6, lines 236-237 is too strongly worded given the stated limitations of the study, and the use of "stage" is again confused for "grade".  Based upon the correlation of positive immunoreactivity and increasing histologic grade in this study, CTEN may be involved in tumor progression.

Author Response

1- In the "Simple Summary", "Abstract", and within the body of the manuscript (page 7, line 244) the authors at times state that there is a correlation with CTEN positive staining and advanced stage of the tumor, when in fact it is a correlation with grade.

The authors corrected the information. In this work, we were not able to assess whether there is a relationship with stage. The authors added the information.

2- Is there further information on the clinical outcome of these patients, especially as correlated to the grading system?  Or any information regarding clinical staging at time of histopathology?  If not, I would explicitly state this in the manuscript.

Unfortunately, it was impossible to obtain the animals' clinical follow-up data. We include this clearly in the materials and methods and also in limitations of the study.

3- In Section 3.4 "Immunoreactivity of CTEN" the use of percentages of positive and negative immunoreactivity for each grade does not seem consistent.  Based upon Figure 1 it seems that percentages should be reported for each tumor grade separately (and this would make most sense).  However, in the text it seems as if percentages are listed across all grades.  Please correct and use consistently.  A table would also be helpful to clarify this information.

The authors correct the information

4- The summary statement on page 6, lines 236-237 is too strongly worded given the stated limitations of the study, and the use of "stage" is again confused for "grade".  Based upon the correlation of positive immunoreactivity and increasing histologic grade in this study, CTEN may be involved in tumor progression.

The authors corrected the information.

Round 2

Reviewer 1 Report

The research topic is interesting but still deserves an editorial refinement in the Materials and Methods (2.1 Animals and histopathology) and content in the Discussion (4. Discussion).

2.1 Animals and histopathology - Authors should indicate the histological staining technique used for SCC grading classification (eg Haematoxylin-Eosin?). It is also necessary to indicate the optical microscope, with the technical specifications, and the image capture system.

Line 138 - please indicate the reference WHO volume.

4. Discussion - Authors should review the text contained between lines 309-314. The authors state that the study "elucidates a novel mechanism...", but the studies cited in support were performed in human SCC and therefore not confirmed with. specific studies in the canine species.

The authors cannot support these multiple actions attributed to Tensin 4/CTN expression in dog SCC considering that they have not planned molecular biological studies to test the regulation (down or up-regulation) of all the mentioned proteins.

Line 292 Remove italics from citation number in the square brackets [38]

Line 334 - the sentence is generic and needs more detailed information regarding the identification of the proteins ( e.g Cadherin E and Cadherin N ) involved in the appearance of the motility of neoplastic cells that have lost homotypic adherence.

Line 335 Down regulation of Cadherin E  may be explaine with E-Cadherin to N-Cadherin in EMT switch process.

Reviewer 3 Report

Thank you to the authors for the revised manuscript.  I feel this has very much improved the paper.  My only minor remaining edits are in the final paragraph of the discussion, beginning at line 375 there are several misspellings and typographical errors that need to be corrected prior to publishing.  
